# Efficacy of a Single-Bout of Auditory Feedback Training on Gait Performance and Kinematics in Healthy Young Adults

**DOI:** 10.3390/s24103206

**Published:** 2024-05-18

**Authors:** Yosuke Tomita, Yoshihiro Sekiguchi, Nancy E. Mayo

**Affiliations:** 1Department of Physical Therapy, Faculty of Health Care, Takasaki University of Health and Welfare, Takasaki 370-0033, Japan; tomita-y@takasaki-u.ac.jp (Y.T.); 2032026@takasaki-u.ac.jp (Y.S.); 2School of Physical and Occupational Therapy, Faculty of Medicine, McGill University, Montreal, QC H3G 1Y5, Canada

**Keywords:** feedback, gait, kinematics

## Abstract

This study investigated the immediate effects of auditory feedback training on gait performance and kinematics in 19 healthy young adults, focusing on bilateral changes, despite unilateral training. Baseline and post-training kinematic measurements, as well as the feedback training were performed on a treadmill with a constant velocity. Significant improvements were seen in step length (trained: 590.7 mm to 611.1 mm, 95%CI [7.609, 24.373]; untrained: 591.1 mm to 628.7 mm, 95%CI [10.698, 30.835]), toe clearance (trained: 13.9 mm to 16.5 mm, 95%CI [1.284, 3.503]; untrained: 11.8 mm to 13.7 mm, 95%CI [1.763, 3.612]), ankle dorsiflexion angle at terminal stance (trained: 8.3 deg to 10.5 deg, 95%CI [1.092, 3.319]; untrained: 9.2 deg to 12.0 deg, 95%CI [1.676, 3.573]), hip flexion angular velocity, (trained: −126.5 deg/s to −131.0 deg/s, 95%CI [−9.054, −2.623]; untrained: −130.2 deg/s to −135.3 deg/s, 95%CI [−10.536, −1.675]), ankle angular velocity at terminal stance (trained: −344.7 deg/s to −359.1 deg/s, 95%CI [−47.540, −14.924]; untrained: −340.3 deg/s to −376.9 deg/s, 95%CI [−37.280, −13.166s]), and gastrocnemius EMG activity (trained: 0.60 to 0.66, 95%CI [0.014, 0.258]; untrained: 0.55 to 0.65, 95%CI [0.049, 0.214]). These findings demonstrate the efficacy of auditory feedback training in enhancing key gait parameters, highlighting the bilateral benefits from unilateral training.

## 1. Introduction

The ability to walk efficiently and safely is crucial for performing daily activities, with implications for independence and social participation. A reduced gait performance can lead to a vicious cycle toward the further deterioration of participation and overall well-being among diverse populations, including the elderly [1], individuals with musculoskeletal [2] and neurological conditions [3], and those with chronic conditions affecting mobility [4]. Strategies aimed at preserving or enhancing gait performance are essential in various healthcare and rehabilitation contexts, underscoring the need for effective gait training interventions.

A reduced gait performance is characterized by slowness, shortened-step length, and small toe clearance [5,6]. These alterations in gait can contribute to a higher risk of trips and falls, decreased mobility, and a consequent reduction in the ability to perform daily tasks independently. Addressing these specific aspects of gait deterioration through targeted interventions is critical for restoring functional mobility and enhancing quality of life. Gait performance is closely related to kinematic characteristics, such as joint angle, joint angular velocity, and muscle activity [7,8]. These kinematic parameters are integral to understanding the mechanics of gait and are pivotal in diagnosing and addressing deviations from normal gait patterns. By analyzing these characteristics, clinicians can tailor rehabilitation strategies to individual needs, optimizing outcomes. Among the various elements of gait, a robust initial heel strike with greater ankle angular velocity has proven to be particularly beneficial. This strong initial contact triggers a crucial series of subsequent gait events, including shock absorption and weight transitioning through the gait cycle [9,10]. Conversely, a lack of proper heel strike can interrupt this sequence of events, resulting in less efficient gait patterns and higher energy consumption [11]. Therefore, prioritizing the refinement of heel contact in gait training could play a key role in enhancing gait mechanics and overall efficiency.

In conventional gait training, therapists frequently rely on verbal and visual cues to highlight key aspects of gait, including a strong heel contact. Traditional verbal cues, while useful, often depend on continuous input from therapists and might not deliver the instant specific feedback required for the most effective motor learning. Biofeedback addresses this by providing immediate performance data, which helps learners to fine-tune their movements in real time. Recent research suggests that integrating biofeedback into gait training improves gait performance [12,13,14]. The incorporation of biofeedback into gait training marks a significant progression, potentially enhancing the impact of rehabilitation programs through more accurate and tailored interventions. Previous studies incorporated various sensory modalities for feedback, such as visual, auditory, and haptic information [12,13,14]. Auditory cue is intuitive and only requires a speaker (or any type of sound device) without requiring special equipment (e.g., visual display and/or actuator) to provide feedback, making it feasible in clinical and community settings. There is accumulating evidence for the efficacy of auditory feedback training on gait performance [12]. However, the changes in the gait performance also depend on gait velocity, making it impossible to understand the efficacy of auditory feedback training on gait performance independent of change in gait speed [15,16]. Furthermore, previous studies only investigated the changes in gait performance on the ipsilateral leg [17,18], while gait performance on one side of the leg may influence that on contralateral leg as well.

While biofeedback training has shown promise in enhancing certain gait parameters, the variability in response based on gait speed and the lack of comprehensive bilateral analysis underscore the complexity of gait rehabilitation. These limitations suggest that individualized approaches and more nuanced metrics of success are needed to fully understand and leverage the benefits of auditory feedback in gait training. Therefore, a knowledge gap exists for the mechanisms underlying the efficacy of biofeedback training and by investigating its impact on bilateral gait dynamics.

The objective of this study was to estimate the immediate change in auditory feedback gait training on gait performance and kinematics of both the trained and untrained legs in healthy young adults. We hypothesized that the effects of auditory feedback gait training on a single lower limb would be observed in bilateral lower limb kinematics.

## 2. Materials and Methods

The sample size was pre-determined based on the effect size of 0.77, as reported in a previous study that examined within-subject changes in step length following auditory feedback training [19]. For the sample size calculation, the G*power 3.1 software was used [20]. The result of the sample size calculation (effect size: 0.77; statistical power: 0.8; alpha error probability: 0.05; Wilcoxon signed-rank test) demonstrated that 16 subjects were required.

The study included 19 healthy young adults, who had no history of neurological or musculoskeletal disorders (Table 1). The study was approved by the Ethics Committee of Takasaki University of Health and Welfare (No. 2252). Written informed consent was obtained from all participants. The dominant leg, determined as the lower limb used for kicking a ball, was the right side for all participants. The procedural flow of the study measurement is shown in Figure 1. Initially, the participants performed the 10 m walk test (10MWT) at a comfortable pace on flat ground to establish the treadmill velocity for subsequent measurements and training sessions. The 10MWT was conducted twice, and the average gait velocity was calculated. The participants then underwent a five-minute familiarization session to acclimate to the treadmill walking. This was followed by a baseline measurement of 60 s to assess gait kinematics. After the baseline measurement, the participants engaged in 10 min of feedback training, after which post-training gait kinematics were measured. The treadmill velocity was consistently set to the comfortable gait velocity determined during the 10MWT for the familiarization, baseline, training, and post-training sessions. Finally, a maximal voluntary contraction (MVC) assessment was conducted to record EMG activity during MVC.

### 2.1. Measurement

To standardize footwear and thus minimize the influence of shoe type on gait kinematics, all participants were provided with identically sized shoes from the same model (JOLT 2, Asics, Kobe, Hyogo, Japan). An IMU-based auditory feedback device (Heel2Toe, Physiobiometrics, Montreal, QC, Canada) was affixed to the lateral side of the shoe on the dominant leg (right side for all participants, as illustrated in Figure 2). Gait kinematics during treadmill walking, both at baseline and post-training, were captured using a 12-camera motion capture system (VICON MX, Oxford Metrics, Yarnton, Oxford, UK) operating at a sampling rate of 100 Hz, alongside a wireless EMG system with four sensors (Delsys Trigno, Delsys, Natick, MA, USA) sampling at 1000 Hz. Thirty-nine reflective markers were secured to the participants’ skin using double-sided tape, arranged according to the plug-in gait full body Ai marker set. EMG sensors were symmetrically placed on the gastrocnemius and tibialis anterior muscles of both legs, adhering to the Surface ElectroMyoGraphy for the Non-Invasive Assessment of Muscles (SENIAM) guidelines [21]. Prior to sensor attachment, skin preparation involved hair shaving, cleansing with alcohol, and the application of a skin treatment solution (Skin-pure, Nihon Kohden, Shinjuku, Tokyo, Japan) to decrease skin contact resistance. Recording sessions of 60 s commenced 60 s after the treadmill reached the predetermined velocity. The baseline foot angular velocity at initial contact (IC) was assessed using the auditory feedback device during the baseline measurement phase.

### 2.2. Intervention

The auditory feedback training comprised a brief instruction about the training (approximately 5 min) followed by a 10 min auditory feedback training session, where the treadmill speed was adjusted to the comfortable gait velocity of the participant as determined by the 10MWT. The feedback device provided auditory feedback for each correct step, gauged by the foot angular velocity at IC measured by the IMU sensor equipped with a triaxial accelerometer, gyroscope, and microcontroller. This device, interfaced with a smart phone via Bluetooth, emitted a beep sound when the foot angular velocity at IC (more negative value is better) exceeded a predefined threshold. At IC, ankle planter flexion occurs, with stronger ICs manifesting as more negative foot angular velocity values. The threshold for foot angular velocity was established at minus 100 deg/s of the value recorded during the baseline measurement (Table 1). The device aims to promote the gait cycle with a strong heel stroke to increase the step length and to facilitate a transition posture from a stooped to an upright posture [22,23]. The device also records the time-series data of angular velocity and acceleration data, along with the success rate defined as the ratio of correct steps to total steps taken. The feedback sound volume was maximized throughout the experiment, corresponding to approximately 50 decibels. This volume level was chosen to ensure that participants could clearly perceive the feedback while walking. The experiment was conducted in a quiet setting to minimize external disturbances.

### 2.3. Data Analysis

#### 2.3.1. Gait Cycle Identification

Kinematic data were analyzed using a custom-made analysis program (MATLAB 2017b, MathWorks, Natick, MA, USA). Three-dimensional marker position data were first filtered using a Butterworth low-pass filter (10th order, cut-off frequency of 5 Hz). The start (stance onset) and end (stance offset) of the stance phase were determined by the points at which the velocity of heel and toe markers decreased below or exceeded 250 mm/s, respectively. The validity of these timings was confirmed through visual inspection. Gait cycles were time-normalized from 0 (stance onset) to 100 (subsequent stance onset), using the linspace function of MATLAB. IC and terminal stance (TS) were, respectively, defined as the periods from stance onset to stance onset plus 20% gait cycle and from 20% before stance offset.

#### 2.3.2. Kinematic Parameters

The following kinematic parameters were derived for both the trained and untrained legs from the recorded data: step length, cadence, toe clearance, joint angle, joint angular velocity, and EMG activity (Table 2). Step length was calculated as the anterior–posterior distance between the heel marker of the measured leg and the toe marker of the opposite leg at the IC of the measured leg. Cadence was determined by counting the number of ICs of the measured leg per minute. Toe clearance was calculated as the minimum vertical distance from the floor surface to the toe marker of the measured leg during the swing phase. The joint angles at IC and TS were calculated for hip flexion and ankle dorsiflexion. The hip flexion angle and ankle dorsiflexion angle were defined as the relative sagittal angles between the pelvis and thigh segments and between the shank and foot segments, respectively. The peak joint angles at IC and TS were identified for both hip flexion and ankle dorsiflexion angles. Joint angular velocity was calculated for the hip flexion and ankle dorsiflexion angles, with the hip angular velocity derived from the peak angular velocity during stance and swing phases, and ankle angular velocity derived from the peak angular velocity at IC and TS. EMG raw data were first demeaned and filtered using a Butterworth band-pass filter (4th order, 50–500 Hz). The filtered EMG data were fully rectified and smoothened using a moving average over a 20 ms time window. The smoothened EMG data were then normalized to the maximum EMG value for each muscle during MVC recordings. EMG activities for gastrocnemius and tibialis anterior were derived by identifying the peak value of the normalized EMG data for the recorded muscle during each gait cycle. All kinematic parameters were calculated for each gait cycle and their mean values were used for statistical analyses.

#### 2.3.3. Statistical Analysis

Non-parametric tests were used to account for the relatively small sample size. Descriptive statistics were presented as median and interquartile range. The Wilcoxon signed-rank test was used to evaluate within-subject changes in the kinematic parameters from baseline to post-training. The Hodges–Lehmann estimator was used to calculate the 95% confidence interval for the within-subject changes in kinematic parameters. The effect size *r* was calculated with the following equation:$$r=\frac{Z}{\sqrt{n}},$$
where *Z* represents the standardized test statistics and *n* denotes the total number of observations. Statistical analyses were performed using SPSS ver. 22 (IBM, Armonk, NY, USA), with a significance level of *p* < 0.05.

## 3. Results

The median (IQR) for the gait velocity and threshold angular velocity were 1.3 (0.2) m/s and −340.0 (158) deg/s, respectively. The success rate for the correct step was 71.5 (33.1) %. Changes in kinematics for the trained and untrained sides are shown in Table 3 and Table 4, respectively. For both sides, the step length (Figure 3A and Figure 4A), cadence, and toe clearance showed significant increases with moderate to large effect sizes, ranging from 0.702 to 0.868. The statistical power for our main outcome (step length) was 0.812. The sagittal hip joint angles remained similar between two time points, while the ankle dorsiflexion angle at TS was significantly greater only in the untrained side with a large effect size of 0.868. The angular velocity of hip flexion exhibited significant increases during both the stance and swing phases on both the trained and untrained sides, with effect sizes ranging from moderate to large (0.526 to 0.738). The angular velocity of ankle dorsiflexion exhibited significant increases (greater negative values) at both IC and TS on both the trained and untrained sides (Figure 3B and Figure 4B, respectively), with effect sizes ranging from moderate to large (from 0.526 to 0.738).

The time-series kinematic data from a representative participant are shown in Figure 5. On both the trained and untrained sides, the ankle angular velocity had negative values at IC and TS. The gastrocnemius EMG activity progressively increased during the stance phase, and its peak occurred at approximately 40% of normalized movement time. Post-training measurements revealed elevated peak ankle dorsiflexion angular velocity and gastrocnemius EMG activity compared to baseline. The analysis of the group data indicated significant enhancements in ankle angular velocity at both IC and TS on both sides, with effect sizes ranging from moderate to large (from 0.563 to 0.812). Similarly, the gastrocnemius EMG activity significantly increased on both sides, with effect sizes ranging from moderate to large (from 0.559 to 0.754). None of the kinematic results are different between genders.

## 4. Discussion

The present study aimed to estimate the immediate effects of a single bout of auditory feedback gait training on bilateral gait performance and kinematics in healthy young adults. The results demonstrate enhancements in step length, cadence, toe clearance, ankle dorsiflexion angle, joint angular velocity, and gastrocnemius EMG activity in both the trained and untrained legs following the treadmill-based auditory feedback training.

A key strength of this study lies in its evaluation of the efficacy of auditory feedback training on gait performance and kinematics while maintaining constant gait velocity using a treadmill. This controlled experiment allowed for the examination of gait kinematic changes independent of those associated with increased gait velocity. Previous research in older adults and individuals with neurological conditions showed that auditory feedback training led to improvements in step length and toe clearance [12,24,25]. In these previous studies, however, these improvements were concomitant with increased gait velocity, thus obscuring the extent to which the kinematic changes were velocity-dependent. Our findings indicate that auditory feedback training can improve the step length and toe clearance even with a constant gait velocity, as ensured by treadmill use. This supports earlier findings that demonstrated a proportional relationship between changes in toe clearance and step length with varying gait velocities [23]. Our study result extends the mechanisms involved in the efficacy of auditory feedback training on gait performance and kinematics.

### 4.1. Possible Mechanisms Underlying the Efficacy of Auditory Feedback Training

The increased step length was associated with elevated joint angular velocities at the hip and ankle during the stance phase. This increase in the joint angular velocity is closely related to an enhanced propulsive joint power [26]. Additionally, our EMG analysis revealed augmented muscle activity in the gastrocnemius, a key contributor to propulsive power during gait, following the training. These findings suggest that auditory feedback training not only enhances gastrocnemius EMG activity, but also augments the propulsive power in the lower limbs, a critical factor for improving gait performance [27].

An increase in the step length typically requires adjustments in sagittal joint angles. Our study found a significant increase in the ankle dorsiflexion angle at TS, whereas the hip flexion angle remained unchanged post-training. The baseline hip flexion angle at TS was approximately −20 degrees, possibly nearing its maximum range of motion, leaving little room for further enhancement. Conversely, the baseline ankle dorsiflexion angle, around 8–9 degrees, indicated potential for additional joint motion.

The combination of increased propulsive power at the hip and ankle joints during the stance phase and a greater ankle dorsiflexion angle at TS could be pivotal in the observed increase in step length following auditory feedback training. These kinematic alterations might stem from modulations occurring in the Central Pattern Generator (CPG), which is a key neural mechanism governing human gait. The CPG is a network of neural circuits in the central nervous system, generates rhythmic movements, and coordinates intersegmental movements for locomotion. Evidence suggests that afferent feedback during gait, including Ia feedback associated with hip extension and ankle dorsiflexion at TS, as well as tactile feedback during heel strike (i.e., IC) and the subsequent stance phase, can modulate the descending neural output from the CPG [28,29,30]. In our study, auditory feedback training appeared to encourage a pronounced heel strike, as indicated by increased peak plantarflexion angular velocity and ankle dorsiflexion angle, potentially enhancing afferent input to the CPG. This heightened afferent input could lead to increased motor neuron activity within the CPG, thereby elevating the gastrocnemius EMG activity.

### 4.2. Transfer Effect of Improved Gait Kinematics to the Contralateral Limb

Our study results indicate that kinematic changes occurred in both the trained and untrained lower limbs, despite auditory feedback regarding IC being provided solely on the trained (right) side. This suggests a clear transfer effect of kinematic alterations from the ipsilateral side to the contralateral side during gait, a phenomenon previously observed in split-belt treadmill training [31,32,33,34]. In such training, asymmetrical gait pattern induced by the split-belt treadmill leads to initial asymmetrical ground reaction forces (GRFs) generated by the lower limbs. However, with continued walking, subjects adapt, resulting in a symmetrical generation of GRFs in both limbs [33]. A similar pattern of adaptation is seen with step length, which starts asymmetrically but becomes symmetrical as the training progresses [34]. These transfer effects of gait kinetics and kinematics may stem from the robust interlimb coordination inherent in locomotion, which is supported by the “half-center control” model of the CPG [35]. In this model, each limb movement is adjusted by a separate CPG, with communication facilitated by inhibitory interneurons that project directly to motor neurons on the contralateral side [36]. According to the half-center control principle, an increased proprioceptive input from load receptors due to enhanced IC after audio feedback training on one side could facilitate the extensor motor neuron output on the ipsilateral side [36], as evidenced by the increased gastrocnemius activity during the stance phase (Table 3). This increased extensor motor output inhibits the activity of the ipsilateral flexor during the stance phase [36]. This inhibition of ipsilateral flexor motor neurons leads to increased flexor activities during the contralateral swing phase through inhibitory interneurons [29,36], as indicated by increased hip flexion angular velocity during the swing phase on the contralateral side (Table 4). Therefore, the robust interlimb coordination, underpinned by the half-center control mechanisms within the CPG, may explain the transfer effect of improved gait kinematics to the contralateral limb.

### 4.3. Clinical Implications

The current study highlights several clinical implications. Firstly, enhancements in gait kinematics following auditory feedback training were observed irrespective of changes in gait velocity, illustrating that an improved IC can be achieved through treadmill-based training. The magnitude of improvement in gait performance and kinematics might be more pronounced during ground walking compared to treadmill training, as an increased gait velocity on the ground may offer additional mechanical benefits alongside the changes induced by enhanced IC. Nonetheless, our findings affirm the advantages of auditory feedback training for gait performance and kinematics, even in situations where treadmill walking is necessitated, such as in cases of postural instability. These results prompt further clinical inquiry into the extent of gait kinematic improvement achievable through auditory feedback training combined with Body Weight Support Treadmill Training (BWSTT), a question that warrants exploration in future research.

Another significant finding is the transferability of auditory feedback training benefits from the ipsilateral to the contralateral side. This suggests that, in cases where gait performance is compromised due to unilateral motor impairments (e.g., hemiplegia, unilateral joint arthritis, and surgery), applying auditory feedback training on the contralesional side could enhance gait kinematics on the ipsilesional side. For instance, in scenarios where clients are hesitant to bear weight on the ipsilesional leg, direct instructions to increase weight bearing may not be effective. However, our findings indicate that auditory feedback aimed at enhancing IC on the contralesional leg could implicitly improve gait performance and kinematics on the ipsilesional leg. The potential benefits of auditory feedback training on the contralesional leg for gait performance and kinematics on the ipsilesional leg merit further investigation in future studies.

### 4.4. Limitations

This study has several limitations. Firstly, the design did not include a control group, which would consist of participants engaging in treadmill walking without auditory feedback. However, all participants were experienced with treadmill walking and underwent a period of familiarization prior to the experiment (Figure 1). They first walked without (control condition) and then with feedback (experimental condition). The sensor was placed on the right shoe and kinematic parameters were obtained on both the trained and untrained side, which served as a control condition as well. For these reasons, we abstained from including a separate control group undergoing treadmill walking training without auditory feedback. Furthermore, the power analysis confirmed that the study was adequately powered to address the research question. The second limitation was that we focused solely on the immediate effects of auditory feedback training on gait performance and kinematics, leaving the long-term impacts unclear. Future research should compare the long-term effects of gait training with and without auditory feedback training, especially in individuals with pathological conditions, such as musculoskeletal or neurological disorders. Thirdly, the training was conducted at a single gait velocity, specifically the participants’ comfortable walking pace on the ground. The effectiveness of the training might vary at different gait velocities, a factor that requires further investigation. Fourthly, the generalizability of our findings is restricted to young healthy adults. To broaden the study’s generalizability, the impact of auditory feedback training on gait performance and kinematics should be examined across a more diverse range of populations. Lastly, although our study clarified the kinematic changes observed in both the trained and untrained limbs before and after auditory feedback training, the efficacy of auditory feedback training relative to other intervention modalities remains an open question. To address this point, future research incorporating control groups will be essential for a more comprehensive understanding.

## 5. Conclusions

Auditory feedback training improves gait performance and kinematics independent of changes in gait velocity in healthy young adults, with efficacy observed in both the ipsilateral and contralateral legs. These findings support the efficacy of auditory feedback training to improve heel strike, thereby benefiting gait performance and kinematics.

## Figures and Tables

**Figure 1 sensors-24-03206-f001:**
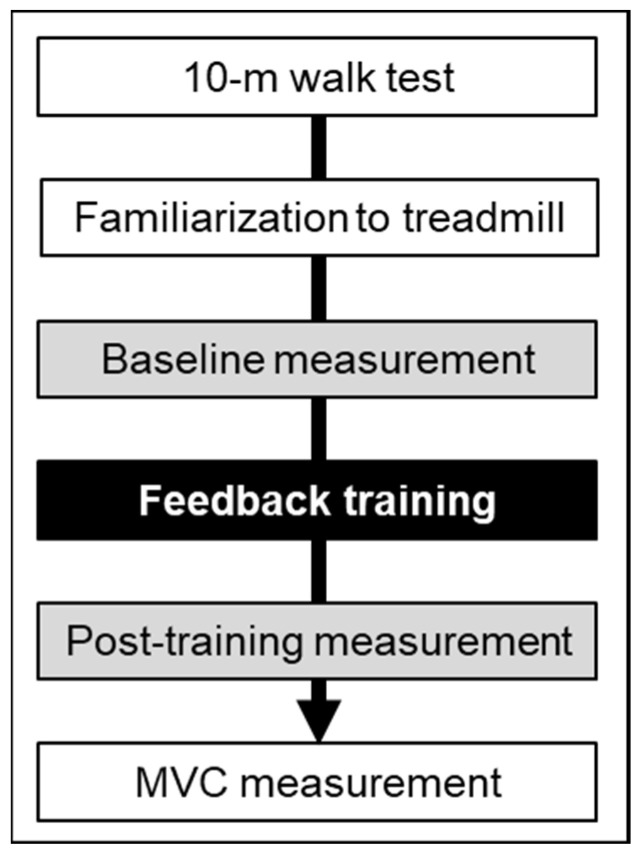
Flow of the experiment.

**Figure 2 sensors-24-03206-f002:**
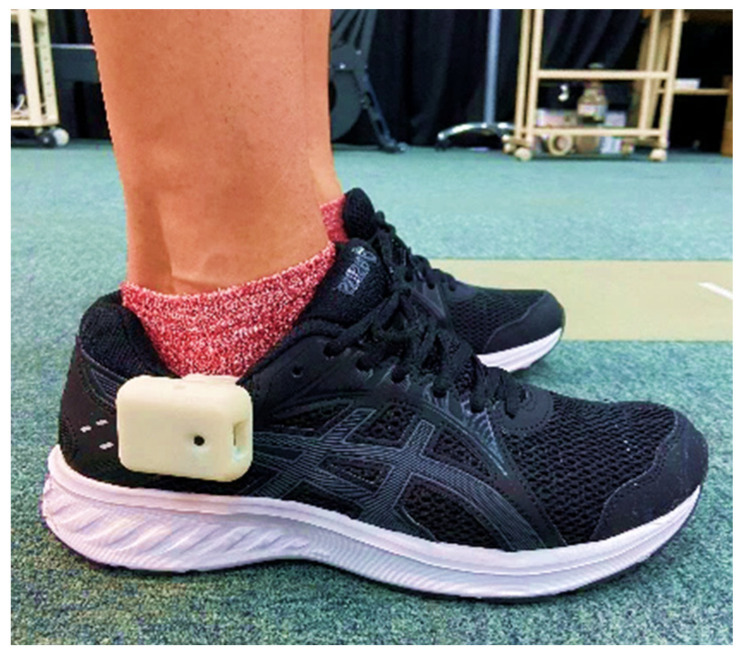
Auditory feedback device was attached on the lateral side of the shoe in the dominant leg.

**Figure 3 sensors-24-03206-f003:**
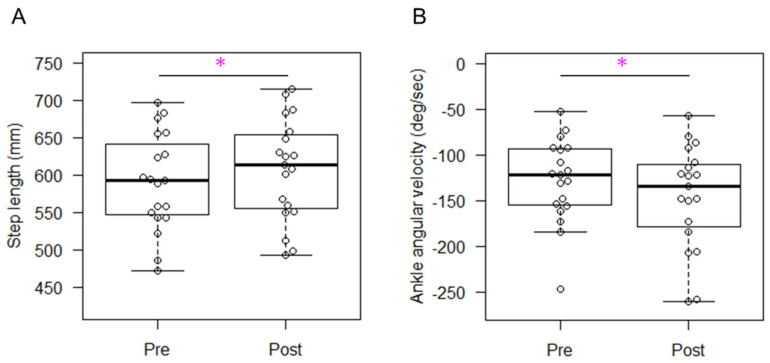
Changes in step length (**A**) and ankle angular velocity at initial contact (**B**) on the trained side. Asterisks indicate statistically significant differences between pre- and post-training.

**Figure 4 sensors-24-03206-f004:**
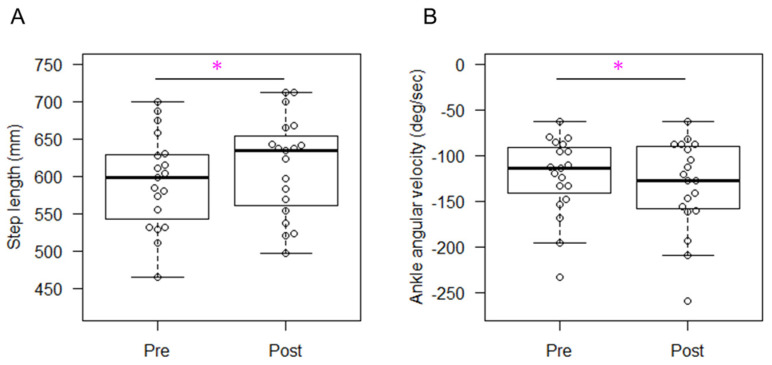
Changes in step length (**A**) and ankle angular velocity at initial contact (**B**) on the untrained side. Asterisks indicate statistically significant differences between pre- and post-training.

**Figure 5 sensors-24-03206-f005:**
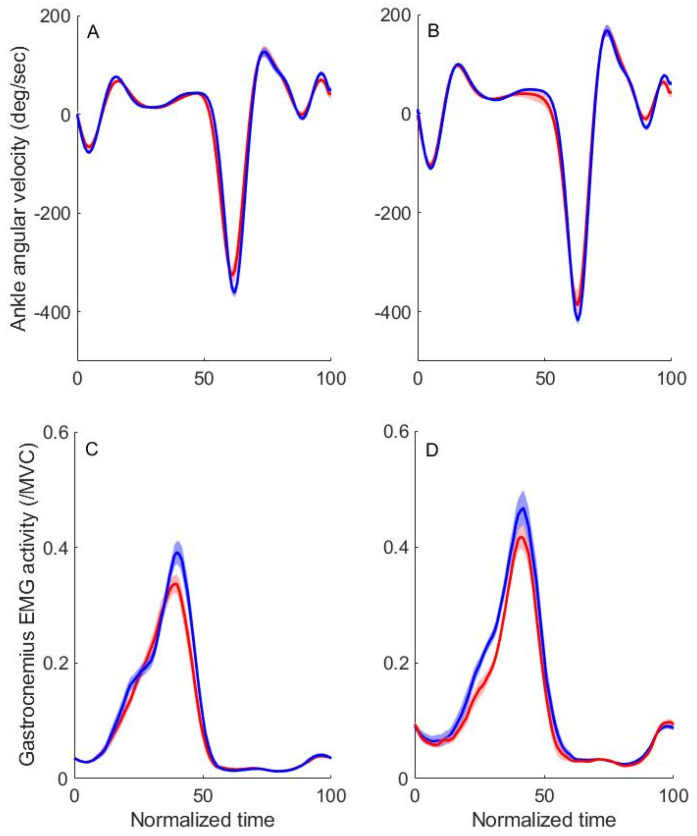
Representative kinematic data in one subject. Kinematic data from one participant are illustrated, showing mean values (thick lines) and 95% confidence intervals (shaded areas) for all steps at baseline (red lines) and post-training (blue lines) across normalized time. Figures display the ankle plantarflexion angular velocity on the trained (**A**) and untrained (**B**) sides, along with the gastrocnemius EMG activity on the trained (**C**) and untrained (**D**) sides. Both peak ankle angular velocity and gastrocnemius EMG activity exhibit increases in the post-training phase (blue lines) relative to baseline (red lines).

**Table 1 sensors-24-03206-t001:** Demographics, median (interquartile range).

Age, years	21.0 (0.7)
Male, n (%)	12 (63.2)
Height, cm	166.4 (1.93)
Body weight, kg	59.6 (1.98)
Gait velocity, m/s	1.3 (0.2)
Threshold angular velocity, deg/s	−340.0 (158)
Success rate, %	71.5 (33.1)

**Table 2 sensors-24-03206-t002:** Definition of the kinematic parameters.

Kinematic Parameters	Unit	Definition
Step length	mm	A–P distance between heel (measured leg) and toe markers (opposite leg) at IC
Cadence	steps/min	The number of IC of the measured leg per minute
Toe clearance	mm	Minimum vertical distance from floor surface to the toe marker of the measured leg during the swing phase
Joint angle	deg	Peak joint angles at IC and TS for both hip flexion and ankle dorsiflexion
Joint angular velocity	deg/s	Peak angular velocity during stance and swing phases (hip flexion) Peak angular velocity during IC and TS (dorsiflexion)
EMG activity	/MVC	Peak EMG activities of gastrocnemius and tibialis anterior during gait

IC: initial contact; TS: terminal stance; EMG: electromyography.

**Table 3 sensors-24-03206-t003:** Changes in gait performance and kinematics in the trained lower limb, median (interquartile range).

	Baseline	Post-Training	*p*-Value	95%CI	Effect Size
**Gait performance**					
Step length, mm	590.7 (91.2)	611.1 (99.9)	0.001	7.609, 24.373	0.729
Cadence, step/min	96.1 (13.2)	93.9 (10.5)	<0.001	−6.521, −0.938	0.822
Toe clearance, mm	13.9 (8.4)	16.5 (9.6)	0.001	1.284, 3.503	0.766
**Joint angle, deg**					
Hip flexion					
Initial contact	20.4 (10.3)	20.1 (10.5)	0.053	−0.009, 1.238	0.443
Terminal stance	−18.0 (10.7)	−18.2 (12.3)	0.494	−0.651, 0.832	0.157
Ankle dorsiflexion					
Initial contact	3.9 (5.9)	4.4 (8.8)	0.872	−1.210, 0.910	0.037
Terminal stance	8.3 (8.5)	10.5 (10.2)	0.006	1.092, 3.319	0.628
**Joint angular velocity, deg/s**					
Hip flexion					
Stance phase	−126.5 (28.0)	−131.0 (17.0)	0.001	−9.054, −2.623	0.738
Swing phase	209.5 (25.9)	224.9 (37.3)	0.008	3.508, 13.635	0.609
Ankle dorsiflexion					
Initial contact	−125.5 (63.1)	−141.2 (86.0)	0.003	−22.772, −3.430	0.674
Terminal stance	−344.7 (105.4)	−359.1 (109.6)	<0.001	−47.540, −14.924	0.812
**EMG activity, /MVC**					
Gastrocnemius	0.60 (0.38)	0.66 (0.57)	0.018	0.014, 0.258	0.559
Tibialis anterior	0.39 (0.24)	0.40 (0.36)	0.420	−0.071, 0.098	0.190

CI: confidence interval (upper limit, lower limit); EMG: electromyography; MVC: maximal voluntary contraction.

**Table 4 sensors-24-03206-t004:** Changes in gait performance and kinematics in the untrained lower limb, median (interquartile range).

	Baseline	Post-Training	*p*-Value	95%CI	Effect Size
**Gait perfomance**					
Step length, mm	591.1 (104.0)	628.7 (100.0)	0.002	10.698, 30.835	0.702
Cadence, step/min	95.9 (11.3)	93.3 (10.7)	<0.001	−6.574, −1.616	0.812
Toe clearance, mm	11.8 (11.6)	13.7 (8.8)	<0.001	1.763, 3.612	0.868
**Joint angle, deg**					
Hip flexion					
Initial contact	22.7 (9.7)	23.8 (27.7)	0.421	−0.408, 0.943	0.185
Terminal stance	−13.0 (11.2)	−13.3 (10.8)	0.658	−0.909, 0.777	0.102
Ankle dorsiflexion					
Initial contact	2.5 (5.9)	3.0 (7.1)	0.601	−0.611, 1.009	0.112
Terminal stance	9.2 (8.9)	12.0 (9.5)	<0.001	1.676, 3.573	0.868
**Joint angular velocity, deg/s**					
Hip flexion					
Stance phase	−130.2 (19.8)	−135.3 (14.1)	0.007	−10.536, −1.675	0.619
Swing phase	216.9 (33.8)	224.8 (35.4)	0.022	1.572, 12.830	0.526
Ankle dorsiflexion					
Initial contact	−117.0 (56.4)	−127.9 (68.7)	0.014	−13.690, −2.700	0.563
Terminal stance	−340.3 (121.6)	−376.9 (105.8)	<0.001	−37.280, −13.166	0.812
**EMG activity, %MVC**					
Gastrocnemius	0.55 (0.23)	0.65 (0.46)	0.001	0.049, 0.214	0.754
Tibialis anterior	0.43 (0.25)	0.45 (0.28)	0.679	−0.116, 0.103	0.098

CI: confidence interval (upper limit, lower limit); EMG: electromyography; MVC: maximal voluntary contraction.

## Data Availability

Data are available from authors upon reasonable request.

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
