# Peer review of "Efficacy of a Single-Bout of Auditory Feedback Training on Gait Performance and Kinematics in Healthy Young Adults"

_sensors, 2024, doi:10.3390/s24103206_

Round 1
Reviewer 1 Report
Comments and Suggestions for Authors
In this paper, authors presented the efficacy of auditory feedback training on gait performance and kinematics in healthy young adults. To further improve the quality of the paper, the following suggestions are put forward:
1. In this paper, there are few analysis diagrams of data characteristics and rules, and only Figure 3 has carried out correlation analysis. Presenting data in tabular form is not as intuitive as a trend chart. Moreover, when Figure 3 is enlarged, it can be found that there are a large number of fuzzy color rectangular color blocks. Is there a formatting or other problem with this diagram?
2. In this paper, auditory feedback is used to optimize gait. What is the volume of the feedback sound in the actual experiment, whether it can be clearly perceived when the human body is moving, and whether it will be weakened by external interference and the sound generated by the movement.
3. The study had a relatively small sample size of 19 healthy young adults. If the authors increase the sample size,the generalizability of the findings could be further improved.
4. The study did not include a control group for comparison.
5. The study focused on the immediate effects of a single-bout of auditory feedback training. Including a long-term follow-up assessment to evaluate the sustainability of the improvements would provide valuable insights into the lasting benefits of the intervention.
Reviewer 2 Report
Comments and Suggestions for Authors
The article is aimed to assessed the efficacy of a single- bout auditory feedback training on gait performance and kinematic in healty young adult that could be an interesting issue, however the research needs additional experiments to demonstrate the efficacy of auditory feedback:
- first of all, a control group should be added to demostrate the efficacy of a training..in the introduction authors declare that "Traditional verbal cues, while useful, often depend on continuous input from therapists and might not deliver the in stant, specific, feedback required for the most effective motor learning.", but there are a extensive research that explain that traditional verbal cue from therapist are efficacy and also the relationship could lead to patients motvation. The authors had to compare "traditional verbal cue" and auditory feedback traning
- the sample is to small to determine the efficacy of auditory training
- a gender stratification may be useful
Round 2
Reviewer 1 Report
Comments and Suggestions for Authors
The authors have carefully modified the content of the paper and the analysis of the results as suggested previously. The research route of the paper is clearer and the research results are more credible.
Author Response
We greatly appreciate the constructive feedback provided in your thorough review. Your insightful comments have significantly enhanced the clarity of our research path and the credibility of our results.
Reviewer 2 Report
Comments and Suggestions for Authors
The research needs additional experiments to demonstrate the efficacy of auditory feedback
Author Response
We appreciate the reviewer's suggestion regarding the inclusion of a control group. Our decision to omit such a group was informed by two key considerations: 1) All participants underwent a familiarization session consisting of a 5-minute treadmill walk (as depicted in Figure 1), which standardized the impact of treadmill walking on gait kinematics for all participants; 2) Our research primarily aimed to evaluate the immediate effects of auditory feedback on gait training, focusing on the performance and kinematics of both trained and untrained legs, rather than comparing outcomes between feedback and no-feedback conditions.
The participants were experienced with treadmill walking and had a period of familiarization prior to the experiment. They walked without the feedback (control situation) and then with feedback. The sensor was placed on the right shoe and kinematic parameters obtained on the trained side and also on the untrained side serving as a control. We did not include the control group undergoing treadmill walking training without auditory feedback in this young, healthy, treadmill-experienced group.
Furthermore, power analysis confirmed that the study was adequately powered to address the specified research question.
We added these points in the Discussion as limitations of the study (Lines 403-412).